# Development of a Vaccine against Human Cytomegalovirus: Advances, Barriers, and Implications for the Clinical Practice

**DOI:** 10.3390/vaccines9060551

**Published:** 2021-05-25

**Authors:** Sara Scarpini, Francesca Morigi, Ludovica Betti, Arianna Dondi, Carlotta Biagi, Marcello Lanari

**Affiliations:** 1Specialty School of Pediatrics, Alma Mater Studiorum, University of Bologna, 40126 Bologna, Italy; sara.scarpini2@studio.unibo.it (S.S.); francesca.morigi@gmail.com (F.M.); ludovicabettimail@gmail.com (L.B.); 2Pediatric Emergency Unit, IRCCS Azienda Ospedaliero-Universitaria di Bologna, 40138 Bologna, Italy; carlotta.biagi@aosp.bo.it (C.B.); marcello.lanari@unibo.it (M.L.)

**Keywords:** hCMV, vaccine, mRNA vaccines, congenital CMV, congenital infection, prevention, CMV long-term sequelae, sensorineural hearing loss

## Abstract

Human cytomegalovirus (hCMV) is one of the most common causes of congenital infection in the post-rubella era, representing a major public health concern. Although most cases are asymptomatic in the neonatal period, congenital CMV (cCMV) disease can result in permanent impairment of cognitive development and represents the leading cause of non-genetic sensorineural hearing loss. Moreover, even if hCMV mostly causes asymptomatic or pauci-symptomatic infections in immunocompetent hosts, it may lead to severe and life-threatening disease in immunocompromised patients. Since immunity reduces the severity of disease, in the last years, the development of an effective and safe hCMV vaccine has been of great interest to pharmacologic researchers. Both hCMV live vaccines—e.g., live-attenuated, chimeric, viral-based—and non-living ones—subunit, RNA-based, virus-like particles, plasmid-based DNA—have been investigated. Encouraging data are emerging from clinical trials, but a hCMV vaccine has not been licensed yet. Major difficulties in the development of a satisfactory vaccine include hCMV’s capacity to evade the immune response, unclear immune correlates for protection, low number of available animal models, and insufficient general awareness. Moreover, there is a need to determine which may be the best target populations for vaccine administration. The aim of the present paper is to examine the status of hCMV vaccines undergoing clinical trials and understand barriers limiting their development.

## 1. Introduction

### 1.1. Epidemiology

Human cytomegalovirus (hCMV), which is one of the eight herpesviruses known to infect humans, is widespread all around the world and is mostly asymptomatic in immunocompetent people [1]. However, hCMV may lead to severe and life-threatening disease in congenitally infected children and immunosuppressed individuals, such as transplant patients and those affected by acquired immunodeficiency syndrome (AIDS) [2].

hCMV is the most frequent cause of vertical transmission infections. The rate of congenital hCMV (cCMV) transmission is between 0.5% and 0.7% of pregnancies in developed nations, and up to 2% of pregnancies in the developing world [3]. The major sources of the virus for expectant mothers are young children with hCMV, most of all toddlers, who can produce saliva and urine with high levels of hCMV [4]. Intrauterine hCMV transmission may occur in mothers without pre-existing antibodies who acquire a primary hCMV infection during pregnancy but also in women who have had previous contacts with hCMV (non-primary infection). In the context of primary infection, 1–4% of seronegative women seroconverted during pregnancy, but the symptoms, when present, are often too mild to seek medical attention. Thus, if hCMV is not tested during pregnancy, the infection is usually not recognized [5,6]. In primary infection, hCMV is transmitted across the placenta in up to 30–50% of the cases, producing fetal infection [5]. Non-primary infection occurs when the fetus is infected because of viral reactivation or in case of maternal reinfection with a different hCMV strain [7]. In such circumstances, the likelihood of hCMV transmission to the fetus is in the range of approximately 3% [5]. However, about three-quarters of cCMV infections are caused by non-primary maternal infection, given the high rates of hCMV seropositivity among women of childbearing age [8]. Vertical transmission is more frequent in mothers with older gestational age, while the risk of fetal damage is higher when infection occurs in the early stages of pregnancy [9,10]. 

hCMV morbidity and mortality can also occur in immunosuppressed people, mostly AIDS or transplant patients, in whom the virus often behaves like an opportunistic pathogen [11,12]. Before the development of antiretroviral therapy, up to 40% of AIDS patients had severe hCMV disease, often sight-threatening [12], while the introduction of such therapy has led to a rapid decline in the incidence of hCMV manifestations and a better prognosis of patients with hCMV disease [13]. In parallel, hCMV infection represents a frequent complication in the setting of hematopoietic stem cell transplantation (HSCT) and solid organ transplantation (SOT) [11], possibly leading to graft rejection. In the case of SOT, when a hCMV seronegative recipient receives an organ from a seropositive donor, the disease develops in more than 50% of cases if no pre-emptive prophylaxis is given [14]. However, even seropositive recipients may have hCMV disease, given the possibility of superinfection with a new strain or reactivation, even if the latter case is less frequent [15]. Differently, after HSCT, the most frequent cause of disease is hCMV reactivation under the influence of immunosuppression in a seropositive recipient [16]. Antiviral prophylaxis, or treatment when needed, is administered routinely to SOT and HSCT patients to prevent serious disease and is significantly but not completely successful [17]. Finally, hCMV may also pose a danger to intensive care unit (ICU) patients, since it has been demonstrated that it is associated with an increased risk of all-cause mortality, increased hospital and ICU length of stay, longer duration of mechanical ventilation, and increased rates of nosocomial infection [18].

### 1.2. Virology 

hCMV is a member of the Betaherpesvirinae sub-family of Herpesviridae [19]. It has an icosahedral capsid that contains the double-stranded DNA, encoding for 165 proteins. As shown in Figure 1, the capsid is surrounded by a tegument made of proteins and, externally, by a lipid envelope [19], whose glycoproteins allow the entry into the host cells through fusion with the cell membrane. After fusion, the viral capsid with the DNA and the tegument proteins are released into the cell [20]. The virus contains various distinct types of glycoprotein complexes (gC) that are useful to infect the host [21]: gC-I is a complex composed of homotrimers of glycoprotein B (gB), a pan-Herpesviridae conserved glycoprotein, mediator of membrane fusion, that rearranges during entry into the cell from a pre-fusion conformation to a post-fusion one; gC-II is the most abundant gC, is made up of glycoprotein M (gM) and N (gN), contributes to the initial binding to the cell membrane, and plays a role in viral replication; gC-III, now called trimer complex (TC), is a heterotrimeric complex where glycoprotein H (gH), L (gL), and O (gO) are linked together, with gH and gL being involved in activating the fusogenic activity of gB, and gO working as a co-receptor. Since its discovery in the middle of the last century, hCMV has been cultivated in fibroblast cell cultures [22]. The discovery of hCMV variants that were no longer able to infect leukocytes and endothelial cells happened at the beginning of the 2000s, leading to the hypothesis that a mutation occurred in laboratory-cultivated strains. This was the starting point of the studies about genetic determinants of hCMV cell tropism [22]. A study carried out by Italian and German researchers in 2002 highlighted the role of the UL128, UL130, and UL131 locus of the hCMV genome in virus growth in endothelial cells and virus transfer to leukocytes [23]. Wang et al. [24] documented the presence of another protein complex, composed of gH/gL combined with UL128, UL130, and UL131 proteins (pUL128–pUL131), different from TC, required to infect epithelial and endothelial cells; they also found that antibodies to either protein of this pentameric structure neutralized the ability of hCMV to infect endothelial and epithelial cells but not fibroblasts. Another study showed that hCMV enters epithelial and endothelial cells by endocytosis followed by low-pH-dependent fusion, which is different from the pH-independent fusion with human fibroblasts [25]. The mechanism was definitively explained in 2008: it was shown that the transition of gH/gL-containing complexes from the endoplasmic reticulum to the Golgi apparatus and cell surface was significantly improved when all three proteins (pUL128–pUL131) were linked to gH/gL, to produce the pentameric complex (PC) gH/gL/pUL128–131 [26]. Another essential step was the discovery of the human receptors of hCMV glycoproteic complexes: platelet-derived growth factor-α receptor, as the cellular receptor of TC [27], and neutropilin2 as the PC receptor in epithelial and endothelial cells [28]. More than half of the proteins produced by hCMV are tegument phosphorylated proteins (pps). After entering the host cell, tegument pps become active, and play important roles in the various stages of the viral life cycle, such as pp65 in immune evasion, pp71 in gene expression, and pp150 and pp28 in virus assembly and egress [19,29]. The expression of herpesviruses genes proceeds with an accurate temporal order consisting of immediate-early (IE), early (E), and late (L) genes [30]. IE genes of hCMV are the first genes expressed after infection and prepare the cell for viral replication [31]. The major immediate-early (MIE) gene seems to have a role in acute infection and is an indicator of viral reactivation. The leading proteins expressed from this region are IE1 and IE2 nuclear phosphoproteins, which regulate transcription and have a role in contrasting the immune response of the host: IE1 antagonizes apoptosis and type I IFN signaling, while IE2 inhibits apoptosis and inflammatory cytokine induction [32]. The role of the various parts of the immune system in controlling hCMV infection has not yet been completely clarified but both innate and adaptive immunity seems to contribute [21].

### 1.3. Clinical Manifestations

In the healthy host, hCMV infection is frequently asymptomatic; rarely, it may present as a febrile illness, or a mononucleosis-like syndrome characterized by fever, lymphadenopathy, and lymphocytosis [1]. Clinical manifestations are overt in congenital disease, transplant, and AIDS patients. In case of cCMV infection, the earliest signs of disease may be seen on fetal anatomy ultrasound at 20 weeks of gestation, even if the sensitivity is lower than 25%. In utero ultrasound indicators include echogenic bowel, brain calcifications and enlarged ventricles, neuronal migration and white matter disorders, microcephaly, fetal growth restriction, oligohydramnios or polyhydramnios, hepatosplenomegaly or hepatic calcifications, ascites, and hydrops [33].

At birth, only 10–15% of newborns with cCMV are symptomatic [34]. The clinical manifestations of cCMV at birth widely vary. Most symptomatic infants are born prematurely, <37 weeks of gestational age [35]. Common manifestations include petechiae, jaundice, hepatosplenomegaly, intrauterine growth restriction, microcephaly [36,37], hypotonia, and seizures, but also abnormal brain imaging findings, sensorineural hearing loss, and chorioretinitis can be associated [37]. Typical laboratory findings are elevated transaminases, thrombocytopenia, and an increase of direct serum bilirubin [35,37]. Permanent sequelae will develop in approximately 45% to 58% of symptomatic newborns and are predominantly motor/cognitive deficits (43%), sensorineural hearing loss (35%), and vision impairment (6%) isolated or in association [38,39,40]. On the other hand, also 10–15% of asymptomatic newborns will have long-term sequelae, mostly auditory disorders [38]. cCMV is the leading cause of non-genetic hearing impairment in children [41], and it is estimated that almost 25% of pediatric hearing loss is associated with cCMV [42]. Indeed, hypoacusia may start within the first years of life, with a median age of onset of 33 months for symptomatic and 44 months for asymptomatic infants [43]. Moreover, it can fluctuate at subsequent examinations, and it may show progressive deterioration [38,44]. The mechanism of hCMV-induced sensorineural hearing loss has not yet been well understood. Huang et al. analyzed the factors that could be responsible for this complication, including the interaction of hCMV with the Wnt and Notch signaling pathways, involved in inner ear development. They highlighted the need to further investigate this issue to provide new directions for therapeutic development [45].

In AIDS patients, the virus causes mostly sight-threatening retinitis that commonly occurs when the CD4+ T-cell count falls below 50 cells/mm3. Less frequently, it is associated with polyradiculopathy, meningoencephalitis, pneumonitis (often in co-infection with *Pneumocystis jirovecii* or *Aspergillus fumigatus*), and gastrointestinal tract infections [46]. 

SOT recipients can develop primary or secondary hCMV infection. Manifestations can vary from myelosuppression, gastrointestinal tract, and central nervous system invasive disease, transplanted organ infection, to acute and chronic allograft injury [47]. Similarly, also in HSCT recipients, hCMV may cause a primary or a non-primary infection. Allogeneic-HSCT recipients are at higher risk than autologous-HSCT ones [48]. Clinical manifestations are similar to those of SOT patients; hCMV pneumonia is frequent in allogeneic HSCT patients, with an incidence rate of 10–30% [49].

### 1.4. Diagnosis

#### 1.4.1. cCMV

Primary maternal infections can be diagnosed by serological tests, considering that the presence of hCMV-specific IgM antibodies indicate acute infection. Since IgM can persist for several months from their appearance, the clinician should test IgG avidity in case of both IgM- and IgG-positive antibodies [50]. Low-to-moderate IgG avidity is typically found for 16–18 weeks following primary infection. Consequently, positive IgM antibodies associated with low IgG avidity are suggestive of infection within the preceding 3 months, while high IgG avidity is indicative of late primary response or non-primary immune response, even if IgM antibodies are also present [51]. Maternal hCMV serology is most useful in the first trimester because of the higher risk of disease in newborns when primary infection occurs in the early stages of pregnancy [52]. However, the majority of cCMV infections are caused by non-primary maternal infection [8]. Universal prenatal screening by testing the serology of pregnant women is not routinely recommended, as there is no currently effective specific treatment to prevent transmission, it is difficult to predict sequelae and an incorrect counseling and interpretation of serology can lead to anxiety, supplementary tests, and unnecessary abortion [53,54]. In case of positive maternal serology, amniocentesis to search for hCMV DNA can be performed [55], because the urine of the infected fetus contained in the amniotic fluid could be positive to hCMV DNA. The sensitivity and specificity of this method reach their maximum when used after 17–20 weeks of gestation and 8 weeks after maternal infection [56]. 

Postnatal diagnosis in newborns is preferably performed via real-time polymerase chain reaction (PCR) on urine or saliva specimens [57,58]. Indeed, infected infants usually eliminate great quantities of virus in saliva and urine; thus, these samples are both suitable for the documentation of cCMV, even if false-positive tests have been reported for saliva [59] and urine collection using a bag may be difficult (e.g., inadequate diuresis, loss of sample, or contamination) [60]. Universal neonatal cCMV screening is not performed, but in the UK, Belgium, Australia, and in some states of the USA, targeted screening of infants who failed the neonatal hearing screening has been trailed [52,61,62]. However, newborns with late-onset hypoacusia would not be identified by this targeted screening. Other methods to diagnose cCMV have been studied, including PCR on dried bloodspot samples that are routinely obtained from newborns for other diagnostic purposes. This technique showed low sensitivity [60]; therefore, it is not suitable for cCMV diagnosis. However, as the dried samples can be stored for several months, it can be used for retrospective investigation of a possible vertical transmission in infants with cCMV infection, although a negative result does not exclude cCMV disease with a delayed onset of signs and symptoms [63]. 

#### 1.4.2. Allogeneic HSCT

hCMV pp65 antigenemia assays and the hCMV DNA PCR are currently the most used laboratory techniques for the detection of hCMV infection in this kind of patient [64,65]. 

#### 1.4.3. SOT

Performing pre-transplantation donor and recipient hCMV IgG serology is recommended, since these serostatuses are key predictors of the risk of hCMV infection after transplant and guide decisions on antiviral prophylaxis or pre-emptive therapy [66]. Serology has no role in the diagnosis of active hCMV replication and disease post-transplantation. Quantitative nucleic acid amplification testing is the preferred method for diagnosis of hCMV infection [66].

### 1.5. Therapy

#### 1.5.1. cCMV

At present, antiviral therapy is only indicated for cCMV symptomatic patients, since there is no evidence of benefit of treatment in asymptomatic neonates. In symptomatic cCMV, antiviral drugs (intravenous (IV) Ganciclovir 6 mg/kg two times a day for 6 weeks or oral Valganciclovir 16 mg/kg two times a day for 6 months) are effective on hearing and neurodevelopmental long-term outcomes [67,68,69,70]. In a retrospective study, Bilavsky et al. observed that infants born with cCMV and hearing impairment receiving Ganciclovir or Valganciclovir for 12 months showed significant improvement in their hearing status [40]. Adverse effects are common in neonates treated for cCMV infection. The main side effect of antiviral therapy is neutropenia, which occurs in approximately 50% of infants and is more common with IV Ganciclovir than with oral Valganciclovir [69,71]. It generally occurs in the first month of treatment, so no increased toxicity was observed in randomized control trials evaluating 6 months vs. 6 weeks of treatment. Neutropenia is rarely severe and usually resolves with dose adjustment, administration of granulocyte colony-stimulating factor (G-CSF), or treatment discontinuation [71,72]. Other common side effects are thrombocytopenia and hepatotoxicity, reported in up to 30% of patients treated with Ganciclovir. Considering antiviral toxicity, it is important to monitor patients regularly with clinical examination and blood tests.

#### 1.5.2. Allogeneic HSCT 

Pre-emptive treatment refers to immediate antiviral administration when hCMV antigenemia or viremia first occurs following transplantation [64]. Ganciclovir is used as pre-emptive therapy in transplant patients, as well as Valganciclovir, Foscarnet, and Cidofovir [64]. Pre-emptive treatment should be maintained until the relevant symptoms are resolved and the hCMV serum test is negative. Subsequently, patients should receive maintenance treatment for a variable period [73]. The length of maintenance treatment varies from 0–6 weeks depending on many factors (such as patients’ sensitivity to treatment, drug side effects, and risk of relapse [74]).

In HSCT symptomatic patients, Ganciclovir is the drug of choice for early treatment but may have a myelosuppressive effect, which can be improved by G-CSF alone or in combination with anti-CMV immunoglobulins. Guidelines recommend induction with IV Gangiclovir 5 mg/kg/die administered twice a day for 7–14 days, followed by maintenance therapy once a day until two consecutive tests are negative [75]. 

Foscarnet demonstrated similar effects to Ganciclovir but did not provoke granulocytopenia, making it suitable for patients with bone marrow suppression [76]. The main adverse reaction is electrolyte imbalance. An additional drug used in pre-emptive treatment is Cidofovir; its administration is weekly, and its main side effect is renal toxicity. Cidofovir is often administered when other treatments have previously been ineffective or in case of intolerance to Ganciclovir or Foscarnet [77]. Letermovir is a novel antiviral drug that suppresses the hCMV-terminase complex [78]. 

#### 1.5.3. SOT

Prophylaxis causes a delayed onset of hCMV disease and reduced complications [79]. Valganciclovir is currently used for this purpose, starting within 10 days after transplant and continuing for 3–12 months [66,80]. In asymptomatic patients, once viremia reaches the positivity threshold, it is recommended to start treatment with Valganciclovir (900 mg every 12 h) and to continue until viremia is negative, with a minimum of 2 weeks of treatment [66]. In case of symptomatic hCMV infection, treatment with Valganciclovir 900 mg every 12 h or IV Ganciclovir 5 mg/kg every 12 h is recommended. This treatment should be continued for a minimum of 2 weeks, until clinical resolution and eradication of hCMV [66].

#### 1.5.4. AIDS

In AIDS patients, antiviral medications against hCMV are used to treat hCMV retinitis: they include systemic (IV Ganciclovir, oral Valganciclovir, IV Foscarnet, or IV Cidofovir) and intravitreal therapy (Ganciclovir, Foscarnet, or Cidofovir) [81].

### 1.6. Why Is It Necessary to Find an Effective Vaccine against hCMV?

hCMV-related disease has a significant impact both on patients’ lives and on health care costs. In transplant populations, morbidity from hCMV is extremely relevant, and antiviral prophylaxis is expensive, not completely successful, and has a limited duration [82]. Every year, thousands of children suffer the consequences of cCMV disease: they are afflicted by permanent disabilities, such as hearing loss, vision loss, and motor and cognitive deficits [83]. Retzler et al. tried to estimate the annual economic burden of managing cCMV and its sequelae in the UK [84]. Their model calculated that the total cost of cCMV to the UK in 2016 was 732 million pounds, 40% of the costs being charged to the public sector while the remaining 60% being indirect costs, such as lost productivity. Moreover, long-term sequelae were reported having a higher financial burden than the acute management. All things considered, a vaccine against hCMV would be highly cost effective [85]. Moreover, hCMV can manipulate the immune system and this seems to compromise self-tolerance in genetically predisposed individuals. It appears that hCMV can lead to immune dysregulation, which triggers the initiation or perpetuation of autoimmune diseases, such as systemic lupus erythematosus and rheumatoid arthritis, with mechanisms, such as molecular mimicry, inflammation, and nonspecific B cell activation [86]. Because of the great commitment required of the immune system to keep the virus silent, there is also growing evidence that hCMV may be the underlying cause of immune senescence and cardiovascular diseases, including atherosclerosis, ischemic heart disease, myocardial infarction, and cardiovascular death [87,88]. Another open issue is the possibility that the presence of hCMV in the host contributes to tumor growth or progression. Recent evidence suggests that hCMV might play a causative role in the pathogenesis and progression of glioblastoma multiforme, colon cancer, and infant leukemia [89].

There is therefore a need to clarify all other possible manifestations and implications of hCMV infection, because this could lead to consideration of universal immunization against hCMV, although this seems a difficult road [90]. Researchers’ and pharmaceutical companies’ attention was brought to this subject in 2000, after the publication of a vaccine priority document written by the National Academy of Medicine of United States that put hCMV vaccine in the group with the highest priority for development [91].

### 1.7. Where Are We Now? 

Despite the clear need, progress toward a vaccine for hCMV has been slow. The first attempts to create a hCMV vaccine began in the 1970s, when two strains of the virus, Towne and AD169, were attenuated in order to function as active immunoprophylaxis [92,93]. The project with the Towne strain was developed by Stanley Plotkin and provided clinical studies with SOT recipients: initial results seemed promising, but statistical analyses revealed that protection against infection was not significant [94,95]. Subsequently, a growing number of hCMV vaccine candidates were developed and are now at different stages, but to date, none have been licensed. There are many intrinsic features of hCMV that are challenging for the design of a vaccine. It can cause persistent asymptomatic infection and establish a lifelong latency in the host after subclinical primary infection [85]; it is able to spread cell–cell, avoiding antibodies in the extracellular fluid [96]; moreover, hCMV reactivations can happen during periods of decreased immune system defenses [85]. Another critical consideration concerns interstrain diversity: hCMV undergoes pervasive recombination with disruptive mutations identified in clinical isolates [97], even with rapid intra-host evolution [98], so that reinfections can occur with different strains [99]. Moreover, hCMV proliferation in the host is species restricted so there are no natural models available to test vaccine strategies [85].

It is well established that children born with congenital infection and immunocompromised subjects are the two groups of patients suffering the most serious consequences after contact with hCMV. Thus, despite the ideal target populations still being controversial at present, it seems reasonable that the more suitable could be pregnant women or women of childbearing age and subjects undergoing transplant [100]. Regarding the vaccination of transplant recipients, the major challenge is inducing an adequate immunity in an immunocompromised individual. It seems that both T cells and neutralizing antibodies are implicated and should be stimulated by a hCMV vaccine in this target population [101]. Concerning cCMV, considering the epidemiological characteristics and modality of vertical transmission, the vaccine should be able to protect both seronegative women from primary infection and seropositive women from reinfection and reactivation [100]. It has been also proposed to include the hCMV vaccine into the routine childhood vaccination schedule, supporting universal vaccination, bearing in mind that the hCMV infection implications are not yet completely clear and that many women are infected by their own children or during jobs that require close contact with children, so the vaccination of toddlers would probably provide indirect protection [82]. With regards to cCMV, we still do not know the exact details of immune mechanisms against this virus, but most data indicate that the vaccine should stimulate both the cellular and humoral components [102]. After a primary maternal hCMV infection in the first trimester, hyperimmunoglobulin administration apparently prevents maternal–fetal transmission, implicitly suggesting a role of antibodies [103], and CD4+ T cells have been correlated with protection against hCMV [104,105]. Considering that infants born to mothers who underwent reinfection or reactivation during pregnancy are still at risk for congenital disease, and that we do not know the specific contributions of humoral and cellular immunity for the prevention of this condition, it could seem very difficult to create an effective vaccine for avoiding vertical transmission [90]. A recent systematic review [5], however, confirmed that the hCMV placental transmission rate is lower in non-primary than in primary infection, with low cCMV infection prevalence in highly seropositive populations. In addition, Tabata et al. [106] showed that human monoclonal antibodies to gB and PC prevent infection in placental cells and anchoring villi better than hyperimmunoglobulin. This evidence might prove that maternal immunity confers protection and pregnant women and women of child-bearing age could be a good target for a hCMV vaccine. Efforts to produce the vaccine have focused on few antigens; neutralizing antibody targets, such as gB, gH, and PC; and T cell epitopes, such as pp65 and IE1 [102]. Initially, gB seemed a perfect choice, but trials showed limited efficacy [102]. At present, not only neutralization but also other antibody mechanisms are known to play a role, such as the induction of complement-mediated virus lysis and antibody-dependent cellular cytotoxicity [102]. A very important step for the development of hCMV vaccine is represented by the identification of new viral glycoproteins and cellular receptors implicated in virus entry in host cells, as previously described [22]. This new awareness led to investigation of the humoral response to hCMV infection, to understand which of the viral antigens could be more important to obtain protective antibodies and to be used in a vaccine [22]. Fouts et al. studied the anti-CMV hyperimmune globulin used to prevent hCMV disease in SOT patients and cCMV and found that the most neutralizing response was provided by antibodies directed against PC, with a little role of anti-gB antibodies [107]. Similar conclusions were also drawn from other studies [108,109], thus supporting the development of PC-based vaccines. To date, the most promising choice in developing hCMV vaccines seems to be an approach that requires the expression of several antigens, with epitopes able to stimulate both the humoral and the cellular components; moreover, the role of protein conformation and structural biology in elicitation of the right immune response has been recognized, as in the case of the pre- and post-fusion crystal structure of gB [102]. Consequently, mRNA and vector vaccines, able to simultaneously synthesize a combination of antigens in their original three-dimensional conformation, could represent good candidates [102]. Chauhan and Singh recently described an immuno-informatics approach to design a multi-epitope vaccine [110]. In the study, a multi-epitope vaccine was constructed thanks to immuno-informatics servers, which selected the most suitable epitopes of PC, gB, and pp65. The results showed that the vaccine might be immunogenic, with high affinity with the immune receptor, effective in stimulating different immune cell types, and could provide long-lasting immune response.

## 2. Current Candidates

In the following paragraphs, we describe the main candidate vaccines developed up to now and the conducted trials; their characteristics are then summarized in Table 1.

### 2.1. Live-Attenuated Vaccines

Live-attenuated vaccines are developed by weakening infectious organisms that can still replicate and induce protective immune responses without causing disease in the host. Because these vaccines are very similar to the natural infection, they usually create a strong and long-lasting immune response.

Live-attenuated vaccines represent the first investigated candidates against hCMV. V160 is a live hCMV vaccine composed by an attenuated AD169 strain where the expression of the PC is restored by passages in endothelial cells [132]. V160 is the first vaccine designed to express the PC, and the results in humans confirm its importance to elicit potent neutralizing titers against viral infection [132,133]. A double-blind, randomized, placebo-controlled phase 1 study was conducted between November 2013 and March 2017 (NCT01986010) [111]. Healthy adults were enrolled and underwent a 3-dose regimen at day 1, month 1, and month 6. This assessment confirmed that V160 vaccine induced immunity, both for the neutralizing antibody and T cell response. Indeed, antibody titers increased with each subsequent vaccination, with 100% of people seropositive at month 7 and a majority still seropositive at month 12 and 18. Furthermore, the V160 vaccine induced a cell-mediated immune response to hCMV, and this increased after every dose. Most people reported only mild systemic adverse events, mostly fatigue and headache. Another similar randomized, double-blind, placebo-controlled phase I study to assess the immunogenicity and safety of the V160 vaccine was conducted and ended in November 2019 (NCT03840174) [112]. It enrolled healthy Japanese males aged between 20 and 64 years, but no results are available yet. Finally, a phase 2 clinical trial is underway (NCT03486834) [113]. In this trial, healthy, hCMV-seronegative women of childbearing age between 16 and 35 years, who have direct exposure to young children at home or occupationally, and who agree to avoid pregnancy during the treatment period, receive 2 or 3 doses of V160 vaccine. They will be followed for 36 months since the first administration, during which the incidence of hCMV infection and of adverse events will be assessed. The study will be completed on July 2021.

### 2.2. Subunit Vaccines

Subunit vaccines only contain the antigenic parts of the pathogen, in particular gB. Because of their limited components and their non-replicating nature, boosted administration and adjuvants are often required to increase the immunogenicity of these vaccines. 

Already in the 1990s, Gonczol et al. described the ability of purified virus envelope and gA/gB complex (made up of glycoprotein gA, gp 55–116, p130/55, gB, and gcI) to induce neutralizing antibodies and a cellular immune response in human volunteers [134]. They immunized two naturally hCMV-seropositive and three hCMV-seronegative human volunteers with gA/gB purified by immuno-adsorbent column chromatography. After a single injection, the naturally seropositive individuals developed higher titers of neutralizing antibodies and temporarily higher hCMV-specific lymphocyte proliferation responses in vitro. The seronegative individuals developed, after the third injection of gA/gB, transiently neutralizing antibodies, and a rapid reappearance and increase in title after the fourth injection. One year after the first injection, the neutralizing antibody titers were still comparable with those of the naturally seropositive individuals. The specific lymphocyte proliferation responses to hCMV in the initially seronegative individuals developed after the second or third injection with the gA/gB preparation and remained positive during the 1-year observation period [134]. This result showed that gA/gB might be used as the basis of a subunit vaccine [134].

Years after this discovery, a phase I randomized, double-blind, placebo-controlled trial with a hCMV vaccine based on gB, combined with an adjuvant, MF59, was conducted [135]. MF59 is a potent adjuvant for human vaccines based on an oil-in-water emulsion of squalene. It is a potent stimulator of cellular and humoral responses to subunit antigens [136]. In this trial, adult participants received hCMV gB vaccine with MF59 or hCMV gB with alum or placebo at 0, 1, and 6 months. A fourth dose was given at 12 months to a specific subgroup. Levels of neutralizing antibody and antibody to gB 2 weeks after the third dose of vaccine were higher than those in seropositive control subjects. The results also showed that formulation with MF59 was more immunogenic than that with alum. The optimal dose of gB was between 5 and 30 µg. Moreover, the fourth dose produced a prompt rise in antibody levels. There were no serious adverse events associated with the vaccine [135].

In the 1990s, other clinical trials demonstrated that the gB/MF59 vaccine was immunogenic and had acceptable profiles of adverse events and side effects. A phase I study investigated the difference in antibody responses between 2 different doses (5 or 30 µg) and 3 different schedules (0, 1, and 2 months; 0, 1, and 4 months; or 0, 1, and 6 months) of hCMV gB vaccine in healthy hCMV-seronegative adults. The vaccine was well tolerated, and there was no significant difference in antibody production between the 2 doses and induced highest antibody titers when given at 0, 1, and 6 months [137]. An open-label (then observer-blinded and randomized) study published in 2002 was conducted on 18 toddlers given either 20 μg of hCMV gB/MF59 or a control hepatitis A vaccine at 0, 1, and 6 months. They concluded that the gB/MF59 vaccine was well-tolerated and highly immunogenic in toddlers [138]. The first phase 2 trial was conducted from August 1999 to January 2010 in hCMV-seronegative women within 1 year after delivery (ClinicalTrials.gov number, NCT00125502) [114]. In this phase 2, placebo-controlled, randomized, double-blind trial, they evaluated a recombinant hCMV envelope gB with MF59 adjuvant vaccine, compared with placebo. Three doses of the hCMV vaccine (234 subjects) or placebo (230 subjects) were given at 0, 1, and 6 months to hCMV-seronegative women. The results showed that the vaccine group had less infections during a 42-month period than the placebo group. Vaccine efficacy was 50%. There were more local and systemic reactions in the vaccine group than in the placebo group. This trial concluded that the gB/MF59 vaccine has the potential to decrease maternal and cCMV infection [114].

In a study published in 2011, the gB/MF59 vaccine was tested in hCMV-seropositive women, to boost the mother’s immunity to hCMV and maybe diminish cCMV infections [139]. This study demonstrated that both CMV-specific antibody and CD4+ T cell responses can be boosted after vaccination with a hCMV gB/MF59 vaccine in women with chronic hCMV infection. Further studies are needed to evaluate whether these boosted responses may prevent vertical transmission of hCMV.

From August 2006 to September 2011, in a phase-2 randomized placebo-controlled trial (NCT00299260), the gB/MF59 vaccine was tested in adults awaiting SOT to evaluate if it could prevent hCMV end-organ disease. They administered either gB/MF59 vaccine or placebo to 70 seronegative and 70 seropositive patients at 0, 1, and 6 months. GB antibody titers were significantly increased in both seronegative and seropositive recipients of the vaccine than the placebo group. In those who developed viremia after transplantation, gB antibody titers correlated inversely with its duration. In seronegative patients with seropositive donors, the duration of viremia and the number of days of ganciclovir treatment were reduced in vaccine recipients [116].

Finally, a randomized, double-blind, placebo-controlled, phase II study assessed the safety and efficacy of the hCMV gB/MF59 vaccine in healthy adolescent females (ClinicalTrials.gov Identifier: NCT00133497). Approximately 400 hCMV-seronegative girls between 12 and 17 years of age received gB/MF59 or placebo at 0, 1, and 6 months. The vaccine induced gB antibodies in all vaccine recipients after 3 doses. Overall, 48 hCMV infections were detected (21 in the vaccine group, 27 in the placebo group). Vaccine efficacy was 43%. They concluded that the gB/MF59 vaccine was safe and immunogenic, although the efficacy did not reach conventional levels of significance and was not sufficient to continue the development of this vaccine [115]. 

### 2.3. Virus Vectored Vaccine

Virus vectored vaccines use a carrier vector to transport hCMV antigens and induce an immune response against hCMV also thanks to the adjuvant effect, which is part of the vector. The presence of pre-existing anti-vector immunity or its potential development could represent a problem for this type of vaccine. 

We describe the hCMV vector-based candidate vaccines tested in clinical trials. Attenuated poxvirus modified Vaccinia Ankara (MVA) is a viral-based vaccine with an ideal safety profile and capable of inducing a strong immune response against recombinant antigens [117]. City of Hope developed a new hCMV vaccine candidate called Triplex by building an MVA encoding three immunodominant hCMV antigens, pp65, IE1-exon4, and IE2-exon5 [117]. Triplex was first evaluated in a phase 1 trial in 24 healthy adults, with or without immunity to hCMV. Three escalating dose levels of the vaccine were evaluated, each one tested in a cohort of 8 subjects who received the vaccine and an identical booster injection 28 days later. hCMV serological status was checked on days 0, 180, and 360. The study established that vaccinations were safe with no dose-limiting toxicities, no serious adverse events, and only mild local and systemic reactogenicity. LaRosa et al. showed significant and long-lasting expansions of hCMV-specific T cells, with potential for viremia control, also in hCMV-seronegative subjects and in adults who previously received smallpox vaccination, thus proving that Triplex is safe and highly immunogenic [117]. 

Consequently, a multicenter, randomized, and placebo-controlled Phase 2 clinical trial was started in 2015, enrolling at-risk, hCMV-seropositive HSCT recipients (NCT02506933). Patients received Triplex or placebo on day 28 and day 56 after HSCT and underwent a 1-year follow up [118]. The primary outcomes were hCMV reactivation, hCMV viremia requiring antiviral treatment, end-organ disease, non-relapse mortality, and severe GVHD. This trial confirmed that Triplex produces hCMV-specific immune responses: the risk for a significant hCMV event during the first 100 days after transplant was reduced by half in patients who received Triplex. Triplex-vaccinated recipients had less hCMV reactivations and higher levels of hCMV-specific T cells than placebo patients. Adverse events were not significantly different between the Triplex and the placebo group [118]. Following the success of this trial, in 2018, City of Hope decided to promote another study, enlisting donors of hCMV-seropositive HSCT recipient (NCT03560752) [119]. This Phase 2 still ongoing trial involves donors receiving one Triplex dose between days 60 and 10 prior to G-CSF mobilization to develop a hCMV-specific T cell response; should this immunity be transferred to the recipients, they might be able to prevent hCMV viremia and reduce the likelihood of hCMV disease before antiviral prophylaxis. Possible administration of the vaccine also to the recipients will be investigated to extend the protection period up to 200 days after transplantation [81,118]. Triplex is currently also under evaluation in the pediatric population: a Phase 1/2 clinical study is proceeding to determine the optimal dose and the protective effect of this vaccine in hCMV-seropositive children receiving an allogeneic HSCT (NCT03354728) [120]. Triplex could also eventually be used in patients who receive SOT and City of Hope is planning studies in this population as well [82,140].

Hookipa Pharma (New York, NY, USA) produced a vaccine called HB-101 using the lymphocytic choriomeningitis virus to express gB and pp65 antigens. This virus is pathogenic to rodents, and it can be used as a vector capable of producing both antibody and cellular responses [82]. In 2016, a placebo controlled, double-blind Phase I trial took place, to evaluate the safety and the immunogenicity of three administrations of HB-101 at three different dose levels in healthy adults (NCT02798692) [121]. HB-101 was well tolerated and induced hCVM-specific cellular responses, mainly pp65-specific CD8 T cell, and neutralizing antibodies production in most subjects. The researchers noted a lack of vector-neutralizing antibody responses, which should help HB-101 functioning [141]. These results led to a randomized, placebo-controlled, Phase 2 trial to assess the safety and efficacy of HB-101 in hCMV-seronegative patients receiving a kidney transplant from seropositive donors. This trial is actually recruiting patients. (NCT03629080) [122]. 

### 2.4. Chimeric Peptidic Vaccines

Chimeric vaccines are recombinant vaccines produced by substituting genes encoding for target antigens from the pathogen in a safe but closely related organism. 

Chimeric vaccines have mostly been studied for the protection of HSCT recipients [142]. The copious tegument protein pp65 has been identified as the major effective antigen to elicit the T cell immune natural response [143]. Indeed, pp65 is one of the main targets for HLA class I-restricted CD8+ cytotoxic T lymphocytes (CTLs). In particular, pp65495-503, a CTL epitope within the pp65 protein that is restricted to the high-frequency HLA A*0201 allele, is considered highly protective due to its low sequence variability among viral isolates and can expand human pp65-specific memory CTLs [144,145]. Cells infected by hCMV express pp65 both early and late after infection, making it an appropriate vaccine target [146]. Furthermore, it was noted that linking pp65 CTL and a T helper epitope markedly enhanced the immunogenicity of these vaccines [147]. Thus, this CTL epitope was fused to a universal T-helper epitope, either a synthetic pan HLA-DR epitope (PADRE) or a natural tetanus (Tet) sequence, with or without CpG 7909, a toll-like receptor 9 (TLR9) synthetic oligonucleotide antagonist, and the fusion showed positive immunogenicity profiles [123]. CpG7909, which is an important receptor expressed in immune system cells, further increases the activity of the vaccine, permitting a reduction of its dosage [123]. Using HLA-restricted CTLs epitopes for a vaccine would eliminate the problems concerning live-attenuated or recombinant live viral vaccines in HSCT recipients. Moreover, it was shown that hCMV complications in HSCT patients were associated with low levels of pp65 CTLs, while protection from reactivation was associated with high levels [148]. Then, a vaccine able to induce protective levels of pp65 CTLs could be of benefit to HSCT recipients [148]. This candidate vaccine was developed by City of Hope National Medical Center and is called CMVPepVax. A non-randomized, open-label, dose-escalating phase 1b safety and immunogenicity clinical trial (NCT00722839) was conducted by City of Hope National Medical Center in California [123]. Healthy adults, aged 18–55 years, molecularly subtyped as HLA A*0201-positive, were enrolled and tested with PADRE-CMV and Tet-CMV peptide vaccines with or without PF03512676 (CpG 7909 adjuvant). Booster doses of vaccine were given at days 21, 42, and 63. The aim of the study was to guarantee enough immunity to HSCT recipients by vaccinating the donors. In fact, several studies showed that it is possible to transfer vaccine-specific immunity from the donors to the recipients [149,150]. In this trial, a pp65-specific vaccine-induced immune response was noticed only when PF03512676 was co-administered, and, in this case, a post-vaccination response was detected in hCMV-seropositive patients, with the immune response before vaccination being minimal to low. In all responders, hCMV immune effects were still detectable at day 77 and persisted until day 180 in 2 subjects who received PADRE-CMV plus PF03512676. Differently, a decrease in hCMV-specific CTLs levels was detected in hCMV-seropositive volunteers who had substantial baseline immune response. A high percentage of flu-like adverse events were reported when using PF03512676. Since the immune response in hCMV-seropositive patients was detected, Nakamura et al. studied the efficacy of PepVax to directly immunize hCMV-seropositive HSCT recipients [124]. A randomized, open-label, phase 1b trial was conducted enrolling hCMV-seropositive patients, aged 18–75 years, positive for HLA-A*0201, who had already undergone HSCT (NCT01588015). Two doses of Tet-CMV vaccine mixed with PF03512676 were administered at day 28 and day 56 after HSCT with good tolerance and did not induce any severe adverse effects or even directly caused acute graft-versus-host disease (GVHD). Compared with the control arm, patients who received the CMVPepVax vaccine had a significantly higher count of pp65-specific CTLs during the first 100 days after HSCT (3.5 vs. 1.4-fold, *p* = 0.025). Moreover, they had less hCMV reactivation (1 vs. 6 events, *p* = 0.039), lower necessity of antiviral use (15 vs. 263 days, *p* = 0.03), and longer relapse-free survival (1 vs. 7 events, *p* = 0.015). Since it has been demonstrated that the median time to hCMV reactivation after HSCT is about 40 days (ranging from 1 to 362), with 98% of reactivations occurring before day 100 [151], this dosing schedule appears to directly target the period of greatest risk for hCMV reactivation. Furthermore, the timing for the first injection seemed adequate, based on safety and favorable clinical outcomes, and it allows patients to have enough time to recover from the acute effects of HSCT. Vaccination at an earlier time could result in low efficacy because of limited haemopoietic reconstitution [152]. Based on these favorable outcomes, CMVPepVax is currently being examined in an ongoing multi-center, placebo-controlled Phase 2 trial (NCT02396134), which will further determine the role and the impact of this strategy in the setting of HSCT [125].

### 2.5. Vaccine Based on Enveloped Virus-Like Particles

This type of vaccine uses supra-molecular protein structures that mimic the real virus but do not contain the viral genome. 

VBI Vaccines Inc. (Cambridge, MA, USA), a biopharmaceutical company, produced a vaccine called VBI-1501 expressing the extracellular domain of hCMV gB fused with the transmembrane and cytoplasmic domains from vesicular stomatitis virus G protein (gB-G eVLPs) [153]. It was evaluated in a Phase 1 randomized, observer-blind, placebo-controlled clinical study (NCT02826798) [126]. The study was designed to assess the safety and immunogenicity of four dose formulations of VBI-1501, with or without alum, in healthy hCMV-seronegative volunteers. Amplification of neutralizing antibody titers was observed after two doses, especially in patients receiving vaccine with adjuvant. Antibodies against fibroblast cells were found in 100% of recipients, and against epithelial cells in 31%. The vaccine was found to be immunogenic at very low doses and there were no security problems [82,126]. A Phase II study had been announced and was expected for the end of 2019, to evaluate how higher doses of VBI-1501 could improve immune responses, but recruitment does not seem to have started yet.

### 2.6. Plasmid-Based DNA Vaccines

Plasmid-based DNA vaccines use plasmids (circular DNA vectors) as versatile platforms in which one or several antigens can be incorporated to induce antigen-specific immunity.

VCL-CB01, also called ASP0113, is a bivalent hCMV DNA vaccine, which contains two plasmids, VCL-6368 and VCL-6365, encoding pp65 and gB; it is combined with two adjuvants, poloxamer CRL1005 and benzalkonium chloride [154,155]. In a phase 1 clinical trial published in 2008, 1- or 5-mg doses of vaccine were administered to 44 healthy adult subjects on a 0-, 2-, and 8-week schedule or 5-mg doses of vaccine on a 0, 3, 7, and 28-day schedule. Overall, the vaccine was well tolerated, with no serious adverse events. Through week 16 of the study, immunogenicity was documented in 45.5% of hCMV-seronegative subjects and in 25% of hCMV-seropositive subjects, and 68.1% of hCMV-seronegative subjects still had memory at week 32 [146].

The second clinical trial of ASP0113 was a multicenter, randomized, double-blind, placebo-controlled phase 2 trial in 108 hCMV-positive, allogeneic HSCT adult recipients (NCT00285259). Subjects were randomized to receive 5-mg doses of the vaccine or placebo prior to ablative conditioning and at approximately 1, 3, and 6 months after transplantation. The vaccine was well-tolerated, and there was a significant reduction in viral load endpoints and increased frequencies of pp65-specific interferon-γ-producing T cells in vaccine recipients compared to placebo recipients; however, there was a lack of significant reduction in the need for hCMV antiviral therapy compared with placebo [127]. The results of this trial provided the basis for defining the primary and secondary endpoints of a global phase 3 trial in HSCT recipients.

The design of a global phase 3 trial (HELIOS) of ASP0113, sponsored by Astellas Pharma Global Development, Inc. (Tokyo, Japan) with Vical as collaborators, was registered at www.clinicaltrials.gov (NCT01877655) [129]. It was a randomized, double-blind, placebo-controlled trial to study ASP0113, in hCMV-seropositive recipients undergoing allogeneic HSCT. Five hundred participants were randomized 1:1 to receive 5 mg of ASP0113 or placebo on days −14 to −3 pretransplant, 14 to 40, 60, 90, and 180 in relation to the day of transplant (day 0). The primary efficacy endpoint of this trial was overall mortality at one-year post-transplantation. The safety of ASP0113 in HCT recipients was also monitored. Secondary outcomes were the percentage of participants with hCMV viremia and with need for hCMV-specific antiviral therapy through 1 year post-transplant. However, ASP0113 did not meet its primary or secondary endpoints in the Phase 3 HELIOS clinical trial, since results did not demonstrate a significant improvement in overall survival and reduction in hCMV end-organ disease. The vaccine was generally well tolerated, with injection-site reactions being the most commonly reported adverse event.

In a phase 2, randomized, double-blind, placebo-controlled study [128] (NCT01974206), conducted from November 2013 to November 2020, the efficacy, safety, and immunogenicity of ASP0113 was assessed in hCMV-seronegative kidney transplant recipients from hCMV-seropositive donors. In total, 150 transplant recipients were randomized (1:1) to receive 5 doses of ASP0113 (5 mg) or placebo on days 30, 60, 90, 120, and 180 post-transplant; they also received prophylactic valganciclovir/ganciclovir 10–100 days post-transplant. The primary endpoint was the proportion of transplant recipients with hCMV viremia ≥1000 IU/mL from day 100 through to 1 year after the first vaccine injection. There was no statistically significant difference in the primary endpoint between the ASP0113 and placebo groups, so it did not demonstrate efficacy in the prevention of hCMV viremia [156].

Only one phase 1, single-center, randomized, open-label trial (NCT00373412) [157] was conducted with VCL-CT02, a hCMV immunotherapeutic trivalent plasmid DNA-based vaccine. It was administered at days 1, 7, and 14, followed by Towne hCMV Vaccine (Towne). Twelve healthy, hCMV-seronegative subjects were enrolled and randomized to receive either VCL CT02 followed by Towne or Towne alone. Safety was monitored and both antibody to hCMV gB and T cell responses to hCMV antigens were measured at specified intervals for 252 days post Towne challenge. The results have not yet been provided.

### 2.7. RNA-Based Vaccines

An RNA vaccine uses a synthetic copy of a natural pathogen messenger RNA (mRNA) and leads the immune system to produce responses against his correspondent antigen. RNA vaccines can be formed by non-replicating mRNA, which only encodes antigens of interest, or by self-amplifying mRNA, which also encodes proteins required for RNA replication. The delivery of mRNA is achieved by a co-formulation of the molecule into lipid nanoparticles, which protect the RNA strands and helps their absorption into the cells.

Several preclinical studies for hCMV RNA-based vaccines have been carried out with excellent results. A major benefit of this technique is the possibility to eventually encode any antigen and to deliver multiple antigens in a single immunization [158]. The use of an RNA vaccine platform offers new opportunities for the development of effective vaccines for a broad range of human diseases, including coronavirus disease 2019 [159]. The first synthetic, self-amplifying mRNA vaccine that was developed contained gB and a pp65-IE1 fusion antigen, both prepared in a cationic nanoemulsion delivery system. After two intra-muscular doses, Brito and colleagues showed that all mice, rats, rabbits, and non-human tested primates developed a potent antibody and T cell response [160]. The second candidate that was tested in a preclinical study contained modified mRNAs encoding gB and PC, both encapsulated in lipidic nanoparticles. The immunization of mice and non-human primates elicited potent and durable neutralizing antibody titers. In order to study the hCMV T cell response, another mRNA vaccine expressing the immuno-dominant hCMV T cell antigen pp65 was developed. In mice, administration of pp65 vaccine together with PC and gB induced a robust T cell-specific response that was highest in case of sequential administration of pp65 alone followed by vaccination with PC + gB + pp65 [161]. A phase I randomized, placebo-controlled, dose-ranging trial sponsored by Moderna evaluated hCMV vaccines mRNA-1647 and mRNA-1443 (NCT03382405) [130]. The first one is composed of mRNAs encoding gB (1 mRNA) and PC (5 mRNAs), while the second contained mRNAs for pp65. They tested the safety and capacity to elicit effective antibody and T cell response in adults aged 18–49 years. In March 2020, the company declared positive seven-month interim safety and immunogenicity data after the third vaccination with mRNA-1647 in the 30-, 90-, and 180-μg-dose cohorts. On the basis of these optimistic results, Moderna built a phase 2 study assessing the safety and efficacy of mRNA-1647 in 252 either hCMV-seronegative or -seropositive healthy adults (NCT04232280) [131]. The vaccine is administered at three dose levels (50, 100, and 150 μg) in three shots (0, 2, and 6 months). The same company is ready to open a global, randomized, observer-blind, placebo-controlled Phase 3 study to evaluate the efficacy of mRNA-1647 against primary hCMV infection in woman of childbearing age [162].

### 2.8. Peptide Vaccines

Peptide vaccines use short peptide fragments to induce highly targeted immune responses. 

In a recent prospective phase I trial (CMVPepVac study: RCHD-CMV-1001, EudraCT No. 2012-002486-35; ISRCTN11842403), a CMVpp65 peptide vaccine was tested in 10 hCMV-negative end-stage renal disease patients prior to renal transplantation. The highly immunogenic nonamer peptide NLVPMVATV derived from hCMVpp65 in a water-in-oil emulsion (Montanide™) plus imiquimod (Aldara™) as an adjuvant was administered subcutaneously four times biweekly. Immunological responses and the clinical course were monitored. This vaccination was well tolerated, and no serious adverse events were detected except for local skin reactions. Fifty percent of the patients had an immune response and 40% presented hCMV-specific CD8+ T cell responses. All responders did not experience hCMV reactivation in the 18 months after transplantation, while all non-responders reactivated. Because of the small numbers of patients, the correlation between hCMV-specific T cell reactivity and vaccine response requires further multi-center studies with larger patient cohorts [163].

## 3. Conclusions

The creation of a vaccine against hCMV is nowadays considered a priority by the international scientific community, considering the emerging data on resistance to ganciclovir, the fact that it would be highly cost effective, and that it could avoid many serious and disabling sequelae of hCMV infection, especially in newborns and immunocompromised patients. However, there is still a need to determine which may be the best target populations to which the vaccine should be administered: pregnant women, women of childbearing age, and young children are those under evaluation. The most promising avenue seems the expression of several antigens within a single vaccine, which could be guaranteed by vector vaccines and mRNA vaccines, able to stimulate more areas of the immune system. The success of vaccines manufactured to combat the SARS-CoV2 pandemic could be an excellent driving force for the definitive development of a vaccine against hCMV.

## Figures and Tables

**Figure 1 vaccines-09-00551-f001:**
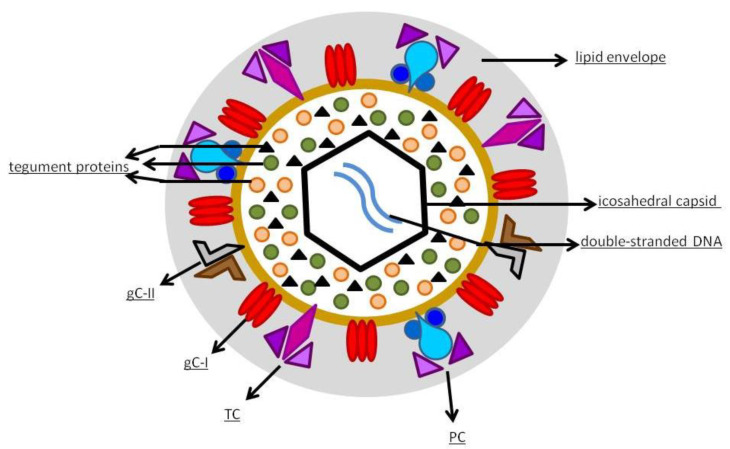
hCMV structure.

**Table 1 vaccines-09-00551-t001:** Overview of the CMV vaccines under clinical development. PC: pentameric complex; NA: not available; gB: glycoprotein B; SOT: solid organ transplant; HSCT: hematopoietic stem cell transplantation; MVA: Modified Vaccinia Ankara; Pp65: phosphorylated proteins 65; IE1: immediate-early 1; IE2: immediate-early 2; Ab: antibody.

Name	Type of Vaccine	Target Site or Antigen	Study Population	Phase, Ref.	Study ID (NCT)	Enrolment Time	Summary of Results
			190 healthy adults older than 18 years: 95 hCMV-seropositive and 95 hCMV-seronegative	Phase I [111]	NCT01986010	November 2013–March 2017	Acceptable safety profile. Levels of antibodies and T cell responses in hCMV-seronegative individuals were within ranges observed after natural CMV infection.
**V160** **Merck Sharp & Dohme Corp**	Live-attenuated	AD169 genetically engineered to express PC	18 healthy males (20–64 years aged), hCMV-seropositive and seronegative	Phase I [112]	NCT03840174	March 2019–November 2019	NA
			2220 females of childbearing age (16–35 years aged), hCMV seronegative	Phase II [113]	NCT03486834	April 2018–ongoing	NA
			464 hCMV-seronegative women within 1 year after delivery	Phase II [114]	NCT00125502	August 1999–January 2010	Vaccine efficacy: 50%; more local reactions and systemic reactions in the vaccine group than in the placebo group
**CMV gB/MF59** **Sanofi Pasteur**	Recombinant subunit	gB with MF59	409 hCMV seronegative adolescent females	Phase II [115]	NCT00133497	June 2006–June 2013	Vaccine efficacy 43%. Safe and immunogenic (although no conventional levels of significance)
			140 adults waiting for SOT (>18 years of age)	Phase II [116]	NCT00299260	August 2006–September 2011	gB antibody titers significantly increased in vaccine than the placebo group
**CMV-MVA Triplex** **City of Hope**	Virus vectored (MVA)	pp65, IE1-exon4, IE2-exon5	24 healthy adults (18–60 years), hCMV-seropositive and seronegative	Phase I [117]	NA	NA	Well tolerated with no dose-limiting toxicities; elicit expansions of hCMV-specific T cells, also in hCMV-seronegative subjects and in adults who previous received smallpox vaccination
			102 hCMV-seropositive HSCT recipients at high riskfor hCMV reactivation	Phase II [118]	NCT02506933	July 2015–January 2021	The risk for a significant hCMV event during the first 100 days after HSCT was reduced by half; less hCMV reactivations and higher levels of hCMV-specific T cells; no significant adverse event
			36 donors of hCMV seropositive HSCT recipient	Phase II [119]	NCT03560752	June 2018–ongoing	NA
			80 hCMV seropositive children receiving an allogeneic HSCT	Phase I/II [120]	NCT03354728	May 2018–ongoing	NA
**HB-101** **Hookipa Biotech GmbH**	Virus vectored (dr)	gB and pp65	54 healthy adults (18–45 years) hCMV seronegative	Phase I [121]	NCT02798692	June 2016–March 2018	Well tolerated; induced hCVM-specific cellular responses, principally pp65 specific CD8 T cell, and neutralizing Ab production
			150 hCMV seronegative recipient awaiting kidney transplantation from hCMV seropositive donors	Phase II [122]	NCT03629080	December 2018–ongoing	NA
		pp65 fused to either pan DR helper T lymphocyte epitope or natural tetanus sequence	68 healthy adults (18–55 years), HLA A*0201 subtyped, hCMV seropositive or seronegative	Phase I [123]	NCT00722839	December 2006–April 2012	No serious adverse events. Immune responses were detected in hCMV-seropositive subjects who received the vaccine co-administered with PF03512676.
**CMVPepVax** **City of Hope, National Cancer Institute**	Chimeric peptidic	pp65 fused to a natural tetanus sequence	36 patients (18–75 years), HLA A*0201 subtyped, hCMV seropositive, who undergone HSCT	Phase Ib [124]	NCT01588015	August 2012–November 2014	Acceptable safety profile. Patients allocated the vaccine had less hCMV reactivation, lower necessity of antiviral use, and better relapse-free survival.
		pp65 fused to a natural tetanus sequence	133 patients (18–75 years), HLA A*0201 subtyped, hCMV seropositive, post-HSCT	Phase II [125]	NCT02396134	May 2015–May 2019	NA
**VBI-1501** **VBI Laboratories**	Enveloped virus-like particles	gB	125 healthy adults (18–40 years) hCMV seronegative	Phase I [126]	NCT02826798	June 2016–August 2017	Immunogenic at very low doses; amplification of neutralizing Ab titers; no safety problems;
			108 hCMV-positive, allogeneic HSCT adult recipients	Phase II [127]	NCT00285259	January 2006–November 2010	Well-tolerated, significant reduction in viral load endpoints, no significant reduction in the need of hCMV antiviral therapy
**ASP0113 (VCL-CB01)** **Astellas**	Plasmid-based	gB, pp65 with CRL1005 and benzalkonium chloride	150 hCMV-seronegative kidney transplant recipients from hCMV-seropositive donors	Phase II [128]	NCT01974206	November 2013–November 2020	No statistically significant difference in the primary endpoint between the ASP0113 and placebo groups.
			514 hCMV-seropositive recipients undergoing allogeneic HSCT	Phase III [129]	NCT01877655	September 2013–September 2017	No significant improvement in overall survival and reduction in hCMV end-organ disease. Well tolerated
**mRNA-1647** **Moderna**	mRNA	mRNA-1647:gB and PC; mRNA-1443: pp65	181 healthy adults (18–49 years), hCMV seropositive and seronegative	Phase I [130]	NCT03382405	November 2017–October 2020	Positive seven-month interim safety and immunogenicity data after the third vaccination with mRNA-1647
		mRNA-1647: gB and PC	452 healthy adults (18–40 years), hCMV seropositive and seronegative	Phase II [131]	NCT04232280	December 2019–ongoing	NA

## Data Availability

Not applicable.

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
