# Peer review of "Development of a Vaccine against Human Cytomegalovirus: Advances, Barriers, and Implications for the Clinical Practice"

_vaccines, 2021, doi:10.3390/vaccines9060551_

Round 1
Reviewer 1 Report
Development of a vaccine against human Cytomegalovirus: advances, barriers and implications for the clinical practice.
This is an excellent review article on vaccines for hCMV. The manuscript was presented in an orderly and comprehensive fashion. The authors presented a fairly good overview of hCMV, epidemiology of the virus, and clinical manifestations of the virus before discussing the present status of hCMV vaccines. Their description and discussion of hCMV vaccines was thorough and detailed.
There are some minor issues with English language usage. A few editing suggestions as follows:
Line 14: if should be in
Line 38: remove paragraph return
Line 53: add a paragraph return after (9, 10).
Line 69: remove paragraph return.
Line 377: The table is listed as both Table 2 and Table 1.
Author Response
R. This is an excellent review article on vaccines for hCMV. The manuscript was presented in an orderly and comprehensive fashion. The authors presented a fairly good overview of hCMV, epidemiology of the virus, and clinical manifestations of the virus before discussing the present status of hCMV vaccines. Their description and discussion of hCMV vaccines was thorough and detailed.
Reply: We thank the reviewer for their positive comments about our paper.
R. There are some minor issues with English language usage.
Reply: we checked English language again and made some corrections.
R. A few editing suggestions as follows:
Line 14: if should be in
Line 38: remove paragraph return
Line 53: add a paragraph return after (9, 10).
Line 69: remove paragraph return.
Line 377: The table is listed as both Table 2 and Table 1.
Reply: Thank you for spotting these inaccuracies, that were amended.
Reviewer 2 Report
The review-article submitted by Sara Scarpini and colleagues entitled “Development of a vaccine against human Cytomegalovirus: advances, barriers and implications for the clinical practice” in times of the coronavirus-pandemic and the common efforts for a vaccine-based “cure” or treatment options captures important clinically relevant considerations and limitations in combating many viral-borne health threats.
Ex ante I would like to point out that indexing of the references is absent- most probably due to an endnote (citation-software) problem- please adjust according to the journal´s style.
The Introduction/Epidemiology section sketches the current knowledge on hCMV prevalence and transmission routes.
The virology section summarizes molecular determinants of this opportunistic DNA-virus for cell-entry, also illustrating the plasticity of the virus to react on the host´s genetic background. In this regard, the disclosure that intense laboratory research (could) drives the virus´ evolution to success is no longer a philosophical question but becomes more and more evident not only to scientists in the current pandemic. The sections clinical manifestations, diagnosis and treatment illustrate that viral-infection symptoms can be recognised by attentive paediatrics and therapeutic/pharmacological interventions should be employed in a reasonable manner. I miss a few words why/how auditory impairments are/might be caused by hCMV- recently reviewed Sheng-Nan Huang et al. in Viruses 2021 (PMID: 33917368).
The sections “Why is it necessary to find an effective vaccine against hCMV?” and “Where are we now” underline the necessity to treat hCMV as infection sequelae are numerous and not only cause a monetary burden to the health system, but to life quality in general. It also becomes clear that a “vaccination of everybody” is not the right direction to “eradicate” the virus, which is anyway not/hardly possible. Rather, vaccination programs should or must be tailored to individuals at higher risk, i.e. children born with congenital infection and immunocompromised people.
The section “current candidates” is summarized in Table 1 and subsequently clinically tested vaccines and outcomes are summarized/indexed in sections live-attenuated-, subunit-, virus vectored-, chimeric peptidic-, enveloped virus-like particles-, plasmid-based DNA- and mRNA- based vaccines. Together, I find these sections satisfactory comprehensive.
Moreover, I find the author´s concluding remarks “there is a need to determine which may be the best target populations to which the vaccine should be administered” very important, a sentence which should/might be appear in the abstract as well.
Author Response
R. The review-article submitted by Sara Scarpini and colleagues entitled “Development of a vaccine against human Cytomegalovirus: advances, barriers and implications for the clinical practice” in times of the coronavirus-pandemic and the common efforts for a vaccine-based “cure” or treatment options captures important clinically relevant considerations and limitations in combating many viral-borne health threats.
Reply: We thank the reviewer for their positive comments about our paper.
R. Ex ante I would like to point out that indexing of the references is absent- most probably due to an endnote (citation-software) problem- please adjust according to the journal´s style.
Reply: We are sorry for the inconvenience. We have corrected this mistake that was actually due to a software problem.
R. The Introduction/Epidemiology section sketches the current knowledge on hCMV prevalence and transmission routes.
The virology section summarizes molecular determinants of this opportunistic DNA-virus for cell-entry, also illustrating the plasticity of the virus to react on the host´s genetic background. In this regard, the disclosure that intense laboratory research (could) drives the virus´ evolution to success is no longer a philosophical question but becomes more and more evident not only to scientists in the current pandemic. The sections clinical manifestations, diagnosis and treatment illustrate that viral-infection symptoms can be recognised by attentive paediatrics and therapeutic/pharmacological interventions should be employed in a reasonable manner. I miss a few words why/how auditory impairments are/might be caused by hCMV- recently reviewed Sheng-Nan Huang et al. in Viruses 2021 (PMID: 33917368).
The sections “Why is it necessary to find an effective vaccine against hCMV?” and “Where are we now” underline the necessity to treat hCMV as infection sequelae are numerous and not only cause a monetary burden to the health system, but to life quality in general. It also becomes clear that a “vaccination of everybody” is not the right direction to “eradicate” the virus, which is anyway not/hardly possible. Rather, vaccination programs should or must be tailored to individuals at higher risk, i.e. children born with congenital infection and immunocompromised people.
The section “current candidates” is summarized in Table 1 and subsequently clinically tested vaccines and outcomes are summarized/indexed in sections live-attenuated-, subunit-, virus vectored-, chimeric peptidic-, enveloped virus-like particles-, plasmid-based DNA- and mRNA- based vaccines. Together, I find these sections satisfactory comprehensive.
Reply: We thank the reviewer for their positive comments about our paper. We added the paper by Huang et al. and some remarks in this regard (page 4).
R. Moreover, I find the author´s concluding remarks “there is a need to determine which may be the best target populations to which the vaccine should be administered” very important, a sentence which should/might be appear in the abstract as well.
Reply: Thank you for your suggestion. We added the sentence to the abstract as proposed.